# Rice Straw Biochar and Magnetic Rice Straw Biochar for Safranin O Adsorption from Aqueous Solution

**Do Thi My Phuong [1,†] and Nguyen Xuan Loc [2,*,†]**

[1] Department of Environmental Engineering, College of the Environment and Natural Resources, Can Tho University, Can Tho 900000, Vietnam; dtmphuong@ctu.edu.vn

[2] Department of Environmental Sciences, College of the Environment and Natural Resources, Can Tho University, Can Tho 900000, Vietnam

[*] Correspondence: nxloc@ctu.edu.vn; Tel.: +84-91-888-9024

[†] These authors contributed equally to this work.

**Abstract:** This study investigates the adsorption of Safranin O (SO) from aqueous solution by both biochar and magnetic biochar derived from rice straw. Rice straw biochar (RSB) was made by pyrolysis in a furnace at 500 °C, using a heating rate of 10 °C·min⁻¹ for 2 h in an oxygen-limited environment, whilst the magnetic rice straw biochar (MRSB) was produced via the chemical precipitation of $Fe^{2+}$ and $Fe^{3+}$. The physicochemical properties of the synthesized biochars were characterized using SEM, SEM- EDX, XRD, FTIR techniques, and $N_2$ adsorption (77 K) and $pH_{pzc}$ measurements. Batch adsorption experiments were used to explore the effect of pH, biochar dosage, kinetics, and isotherms on the adsorption of SO. Experimental data of RSB and MRSB fit well into both Langmuir and Freundlich isotherm models, and were also well-explained by the Lagergren pseudo-second-order kinetic model. The maximum SO adsorption capacity of MRSB was found to be 41.59 mg/g, while for RSB the figure was 31.06 mg/g. The intra-particle diffusion model indicated that the intra-particle diffusion may not be the only rate-limiting step. The collective physical and chemical forces account for the adsorption mechanism of SO molecules by both RSB and MRSB adsorbents. The obtained results demonstrated that the magnetic biochar can partially enhance the SO adsorption capacity of its precursor biochar and also be easily separated from the solution by using an external magnet.

**Keywords:** adsorption; biochar; magnetic biochar; Safranin O; rice straw; aqueous solution

## 1. Introduction

The manufacture of textiles, leather, paper, pharmaceuticals, food, synthetic rubber, plastics, and paints produces a considerable amount of wastewater. Typically, this water contains dye effluents and is directly discharged into the environment without treatment. Dyes can be of many different structural varieties, generally divided into cationic, nonionic, or anionic types [1]. Safranin O (SO) is a red cationic dye and belongs to the azine group, which is extensively used in histology, cytology, bacteriology, and so forth. Safranin O has a complex organic structure and is frequently found in trace amounts in the industrial discharge waters. The release of synthetic dyes such as SO into the water environment not only obstructs light penetration into the surface waterbodies, thus reducing photosynthetic activity and endangering aquatic life, but is also toxic to aquatic microorganisms and seriously affects human health if not treated properly [2].

Conventional wastewater treatment removes coloration through membrane filtration, coagulation-flocculation, advanced oxidation, electrochemical methods, or microbial degradation methods. However, such approaches often have low efficiency, are high-cost, and not environmentally safe [3]. Adsorption has been considered as one of the most effective removal methods for effluents containing synthetic dyes, thanks to its flexibility

and simplicity in design [3]. In recent years, many more studies have focused on adsorption treatment methods by using low-cost adsorbents, such as biochar, instead of activated carbon. Compared to activated carbon, biochar is less energy- and cost-intensive due to lower temperatures used in its production [4].

Biochar is a porous carbon-rich solid product resulting from the thermal degradation (pyrolysis) of lignocellulosic-derived biomass under an almost oxygen-free environment. Biomass waste materials appropriate for biochar production include grass, wood chips, wheat straw, seed husk, and bagasse.

Vietnam, with a large agricultural sector, has significant biomass residue potential for biochar production. Mekong River delta is the key rice-producing region in Vietnam. With an estimated rice production of approximately 43 million tons/year (GSO, 2021), rice straw residues offer a promising source for biochar production.

Biochar has been evaluated as a potential adsorbent for water treatment due to the multi-functional properties of the material. Its high porous structure and stability in water are considered favorable for its adsorption capacity. Additionally, the presence of several functional groups on its surface mainly include carboxyl (–COOH) and hydroxyl (–OH), which can form complexes with many classes of dyes [5]. Several applied studies have investigated the biochar adsorption of dyes in solution [6,7]. However, as with other powdered adsorbents, the powdered biochar after being used would cause secondary pollution in water bodies. This often requires filtration and centrifugation processes in further treatment steps. Difficulty in separating powdered biochar hindered its application on a large scale.

The synthesis of biochar-based magnetic composites is of increasing interest for its properties of being quickly separated from water by using an external magnet. Pyrolysis, coprecipitation, and calcination are the methods frequently used in the preparation of magnetic biochar [8]. Among them, the co-precipitation method is mostly used to synthesize magnetic biochar, as it requires an easier process. The coprecipitation procedure describing the preparation of magnetic biochar from $Fe^{2+}/Fe^{3+}$ salt solution is summarised by the following equation:

$$Fe^{2+} + 2Fe^{3+} + 8OH^- \rightarrow Fe_3O_4\downarrow + 4H_2O.$$

For the magnetization of biochar, different magnetic medium can be used. The majority of previous studies have selected iron-based materials such as $FeCl_3$, $Fe_2O_3$, or $Fe_3O_4$ as the magnetic medium. The evaluation of the characteristics of the magnetic biochar can be determined by X-ray diffraction, Fourier transform infrared spectroscopy, and scanning electron microscopy [9]. Magnetic biochar overcomes problems related to filtration of non-magnetic biochar, as they can be removed from the water after its adsorption by using powdered external magnetic fields. Magnetic biochar applications have already been practiced. For example, the ultrafine magnetic biochar/$Fe_3O_4$ adsorbed 62.7 mg/g of carbamazepine [10]. Another magnetic biochar, produced by the precipitation of strontium hexaferrite ($SrFe_{12}O_{19}$) onto sewage biochar surfaces, exhibited high removal malachite green capacity (388.65 mg/g) [11]. Magnetic biochar from alkali-activated rice straw adsorbed 53.66 mg/g of rhodamine B [12].

Rice straw conversion into biochar and magnetic biochar is a growing field of interest due to a variety of potential applications, including dye adsorption from aqueous solution. Several rice straw precursors have been used to synthesize magnetic biochar-based composites for removal of rhodamine B [12] or crystal violet [13]. It has been reported that with magnetic coating, the adsorption capacity of magnetic biochar to organic pollutants from aqueous solution is increased as compared to their precursor biochar. For example, corncob magnetic biochar produced by using low-temperature hydrothermal methods showed a higher adsorption capacity for Methylene blue to magnetic biochar (163.93 mg/g) than the pristine biochar (103.09 mg/g) [14]. The co-precipitation of $Fe^{2+}/Fe^{3+}$ on orange peel powder has been shown to have adsorption capacities for hydrophobic organic compounds and phosphate than the non-magnetic biochar [15]. Furthermore, corn stalk

biochar coated with magnetic $Fe_3O_4$ nanoparticles by co-precipitation achieved adsorption capacity for crystal violet close to 20 times greater than that of precursor biochar [16]. However, no study has investigated the removal of SO dye using magnetic biochar from rice straw or a comparative evaluation adsorption capacity of SO between magnetic biochar and precursor biochar. Therefore, in this study, an exploration of the physicochemical characteristics in addition to the comparative evaluation of both biochar and magnetic biochar for Safranin O adsorption from solution was examined. Batch experiments were performed using different variables, including pH solution, adsorbent dosage, initial SO concentration, and contact time, whilst the isotherms and kinetics were studied to explore the adsorption mechanism of SO adsorption.

## 2. Materials and Methods

### 2.1. Chemicals

All chemicals including ferric chloride ($FeCl_3 \cdot 6H_2O$), ferrous chloride ($FeCl_2 \cdot 4H_2O$), hydrochloric acid (HCl), and sodium hydroxide (NaOH) were provided by Merck (Darmstadt, Germany). Safranin O (SO) was supplied by Sigma-Aldrich and was used without any pretreatment. The stock solution of SO was prepared (1000 mg/L), and the initial concentration was adjusted to the desired concentration by diluting with deionized distilled water. The chemical structure of SO is demonstrated in Figure 1.

**Figure 1.** Chemical structure of the SO dye.

### 2.2. Preparation of Biochar and Magnetic Biochar from Rice Straw

Biochar preparation: The raw rice straw collected from the Vietnamese Mekong Delta was first air-dried and cut into small pieces of ca. 2–4 mm size, then formed into cylindrical granules. Pyrolysis occurred in a furnace (Model VMF 165, Yamada Denki, Adachi, Tokyo, Japan) at 500 °C, with a heating rate of 10 °C·$min^{-1}$ for 2 h in the absence of oxygen. After cooling, the biochar was crushed and sieved (grain size <0.075 mm). The sieved biochar was then washed with a solution of 0.1 M HCl and distilled water until the pH was between 6.0 and 7.0. The biochar was finally dried overnight at 80 °C and stored until use or further magnification. The samples were identified as rice straw biochar (RSB).

Magnetic biochar preparation: Magnetic rice straw biochar was synthesized using the high-temperature co-precipitation method. Here, biochar was mixed with a suspension of various Fe-chlorides under very alkaline conditions (pH = 10) in order to precipitate Fe-hydroxides, according to protocol summarized by Sun et al. (2015). Thereafter, it was precipitated using $FeCl_3 \cdot 6H_2O$ and $FeCl_2 \cdot 4H_2O$ with a ratio of 3:1, where the pH was adjusted to 10 using a 5 mol/L NaOH solution. The mixture was stirred on a magnetic stirrer with RSB at 80 °C and for 1 h, then was separated by centrifugation at 3000 rpm for 10 min. The collected precipitate was finally oven-dried at 60 °C to constant weight. The samples were identified as magnetic rice straw biochar (MRSB).

### 2.3. Characterization of Adsorbents

A scanning electron microscopy SEM (Hitachi S-4800, Hitachi Ltd. Tokio, Japan) was used to visualize the microstructure of biochar-based adsorbents, coupled with an energy-dispersive X-ray (EDX) spectroscopy (Hitachi, Japan) to analyze the elemental surface composition of the biochar-based adsorbent samples. Fourier transform infrared

spectroscopy FTIR (FTIR-PerkinElmer Spectrum 10.5.2, Buckinghamshire, UK) was used to determine the functional groups on the surface samples. X-ray diffraction (XRD) for crystal phase recognization was performed using Bruker D8 Advance (Bruker AXS GmbH, Karlsruhe, Germany). The BET-specific surface area was calculated on the basis of low-temperature nitrogen adsorption isotherm measured using Nova Station A (Quantachrome Instruments version 11.0, Miami, FL, USA). The detailed sample preparation for SEM, EDX, FTIR, and BET analysis is provided in the Supplementary Material.

*2.4. Batch Adsorption Experiments*

The adsorption performance of RSB and MRSB against SO in solution was evaluated in the batch adsorption experiments. All the adsorption tests were conducted in triplicate and the average results were taken. In a typical experiment, the fixed quantity of adsorbents and 10 mL SO solution were added into #15 mL conical centrifuge tubes. The flasks were agitated in a shaker (HS 250 Basic, IKA Labortechnik, Ho Chi Minh, Vietnam,) at 120 rpm at room condition ($25 \pm 2$ °C) for a fixed time. The solutions were filtered by Whatman No. 6 filter paper (pore size, 3 μm). After filtering, the residual SO in solution was analyzed by measuring the SO at an optimal wavelength of 530 nm (corresponding to the maximum adsorption capacity for SO), using UV-Vis spectroscopy (Shimazdu, Japan).

To determine the optimal experimental conditions, a series of preliminary tests were performed with variation of solution pH (2–10), adsorbent dosage (1–5 g/L), SO concentration (10–200 mg/L), and contact time (1–720 min). The optimum results obtained were then used for kinetic and isotherm studies.

The amount of SO adsorbed ($q_e$, mg/g) was determined according to the following equation:

$$q_e = \frac{C_0 - C_e}{m} V$$

where $C_0$ (mg/L) is the initial dye concentrations; $C_e$ (mg/L) is the dye concentrations at equilibrium; m (g) is the mass of adsorbent; and V (mL) is the volume of the solution.

## 3. Results

*3.1. Characteristics of Adsorbent Materials*

The synthesized RSB and MRSB materials were characterized using the different techniques and measurements, in which the calculation of a BET-specific surface area provides a quantitative estimate of the adsorbent surface area per unit mass for adsorbed molecules. The synthesized magnetic biochar of previous studies exhibited a greater or lesser specific surface area than their precursor biochar [17,18]. In this study, the introduction of a magnetic medium to rice straw biochar improved the specific surface area. In particular, the specific surface area of MRSB was observed to be 337.77 m²/g, while the value for the precusor biochar was 269.93 m²/g (Table 1). This may be due to the additional surface area provided by the available iron oxide particles embedded in the biochar matrix. An increase in surface area of magnetic biochar versus precursor biochar has also been reported in other studies [19,20].

**Table 1.** The surface area, particle size, and pH$_{pzc}$ of RSB and MRSB.

|  | RSB | MRSB |
|---|---|---|
| Specific surface area (m²/g) | 269.93 | 337.77 |
| Average particle size (nm) | 2.26 | 1.74 |
| pH$_{pzc}$ | 6.90 | 7.69 |

The surface charge and chemistry of an adsorbent in water varies significantly with pH, and therefore it is highly relevant to determine the point of zero charge (pH$_{pzc}$). pH$_{pzc}$ is defined as the pH at which the adsorbent material surface is not charged, that is, the net

electrical neutrality. In solutions having a pH less than $pH_{pzc}$, the adsorbent will have a net positive surface charge, for if the pH is greater than $pH_{pzc}$, the surface of the adsorbent will be negatively charged. The $pH_{pzc}$ value of the RSB and MRSB in this study was determined using the "drift method" [21], and their values are presented in Table 1. The $pH_{pzc}$ of MRSB (7.69) was found to be higher than that of RSB (6.90). The $pH_{pzc}$ in MRSB slightly increased, probably due to the presence of the Lewis basic groups (i.e., groups having a –OH bond), after iron oxides were added on the precursor biochar surface [22].

To study the surface morphology of the synthesized materials, SEM images of RSB and MRSB at different magnifications are shown in Figures S1 and S2 of the provided Supplementary Material. The surface of the RSB from the SEM images possessed cylindrical and porous structure forms, while the SEM image recorded for the MRSB resembled flake-like structures. An examination of the EDX element map of RSB and MRSB is represented in Figures 2A,B, respectively, while the summary is shown in the Table presented within these Figures. In the EDX results, the RSB mainly consisted of C, O, Si, and Al. After impregnation of magnetite, there were variations in the percentages of elements C, O, Al, and Si, and the disappearance of P, K, Mg elements, as well as the appearance of large amounts of iron (accounting for 20.73% atomic weight) on the MRSB surface. Precipitated iron oxides formed during $Fe^{2+}/Fe^{3+}/NaOH$ treatment (primarily magnetite) [23]. Therefore, the presence of the Fe element (detected by the SEM-EDX) in MRSB indirectly confirmed the appearance of iron oxides in the magnetic biochar materials.

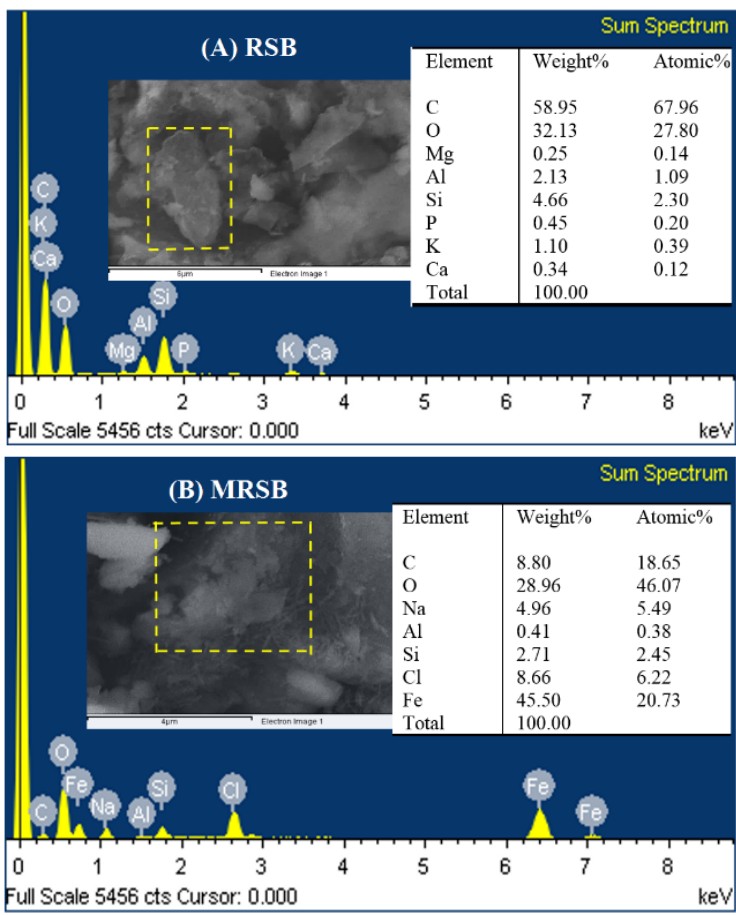

**Figure 2.** SEM images and EDX mapping of RSB (**A**) and MRSB (**B**) surfaces.

Figure 3 shows the X-ray diffraction (XRD) patterns of RSB and MRSB, and the diffraction pattern of RSB did not exhibit any remarkable crystallization peaks. Therefore,

the diffraction patterns of the RSB may be obtained from the amorphous phase. Compared to the RSB diffraction spectrum, the diffraction peaks of MRSB were assigned to (220), (311), (422), (440), and (533) planes (JCPDS No. 19-0629, a = 8.396 Å). These planes can be assigned to the diffraction of $Fe_3O_4$ (magnetite) crystal with an inverse spinel structure [23]. In addition, no other diffraction peaks were observed corresponding to possible impurities. This is indicative of high phase-purity of the products, with magnetite as the main crystallite phase in MRSB.

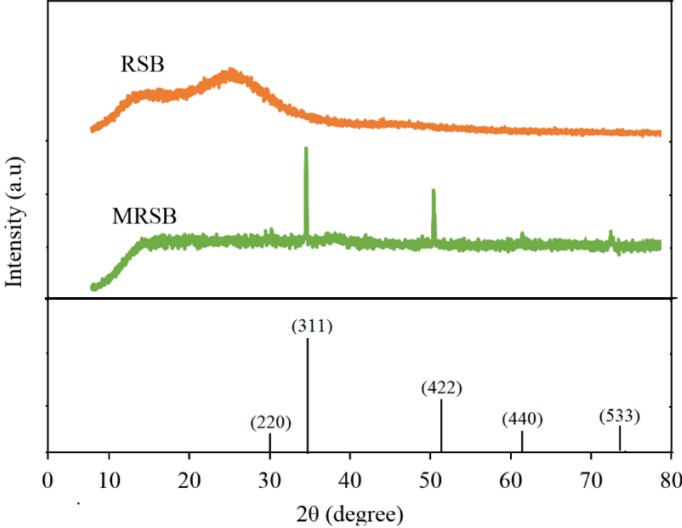

**Figure 3.** XRD patterns of RSB and MRSB.

FTIR of the RSB and MRSB found aromatic C-H, C–O–C, aromatic C-H, C=C, aliphatic C–H and –OH groups stretching at 712–797 cm⁻¹, 1039 cm⁻¹, 1368 cm⁻¹, 1621 cm⁻¹, 2931 cm⁻¹, and 3322 cm⁻¹, respectively (Figure 4). Interestingly, the FTIR of the MRSB appeared to have a new strong absorption peak at the band of 640 cm⁻¹ related to the Fe–O bond in $Fe_3O_4$ [19]. The FTIR results were hereby consistent with the XRD and EDX analysis results, demonstrating that rice straw biochar was successfully magnetized.

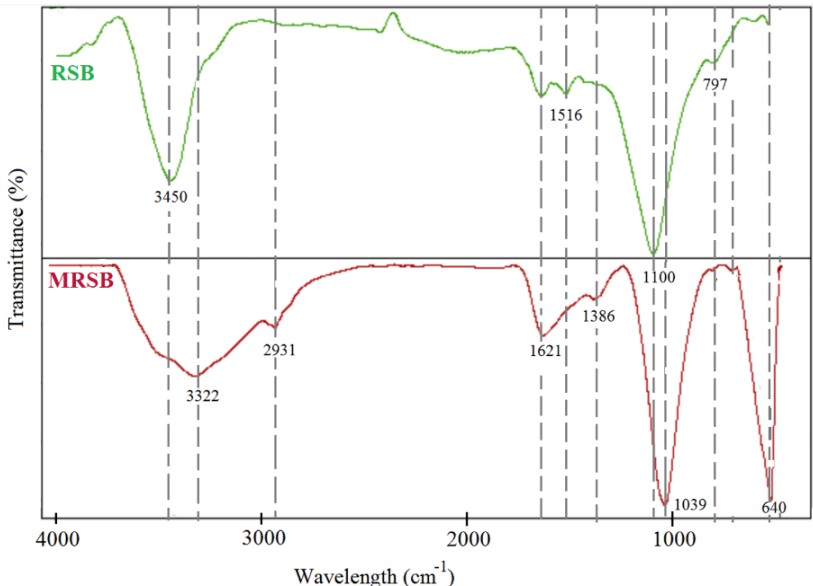

**Figure 4.** FTIR analysis of RSB and MRSB.

Overall, the SEM-EDX, XRD, and FTIR findings demonstrated that the magnetic rice straw biochar MRSB was successfully synthesized and deposited on the surface of the pristine biochar RSB.

It should be noted that the preparative methods for adsorbent materials from rice waste can be scaled up to kilogram levels for commercial-scale production, if processed properly. For the production of biochar and magnetic biochar from other types of biomass sources, new processes should be established based on actual conditions depending on cost-efficiency, including (1) the biomass diversity and their availability in the region; (2) the spread of harmful gas pollutants from combustion treatment to the surrounding environment; (3) the cost of pre-treatment, magnetization, storage and disposal; and (4) the type of process and processing conditions.

### 3.2. Adsorption Isotherm Analysis

To determine the optimum experimental parameters used for kinetic and isotherm studies, preliminary tests were firstly conducted, with varying solutions of pH (2–10, controlled by 0.1 M HCl or 0.1 M NaOH), adsorbent dosage (1–10 g/L), initial SO concentration (10–200 mg/L), and contact time (1–720 min). The obtained optimum results are shown in Figure S3–S6 (Supplementary Material). The optimum conditions for SO adsorption by both adsorbents were determined to be at pH ~7, a SO concentration of 50 mg/L, an adsorbent dosage of 2 g/L, and at an adsorption time of 240 min.

Adsorption isotherms are often used when evaluating the effects of initial contaminant concentrations. However, Langmuir and Freundlich isotherm models are common in wastewater engineering for data analysis, and their non-linear equations are as follows:

$$q_e = \frac{q_m K_L C_e}{1 + K_L C_e}$$

$$q_e = K_F\, C_e^{1/n}$$

where $q_e$ (mg/g) represents the equilibrium adsorption capacity; $q_m$ (mg/g) represents theoretical maximum adsorption capacity; $C_e$ (mg/L) represents the equilibrium concentration of the adsorbate; $K_L$ (L/mg) represents the Langmuir adsorption constant; $K_F$: $((mg/kg)/(mg/L)^n)$ represents the sorption affinity; and $1/n$ represents the nonlinearity index (unitless). Figure 5 shows the fitting result of Langmuir and Freundlich models.

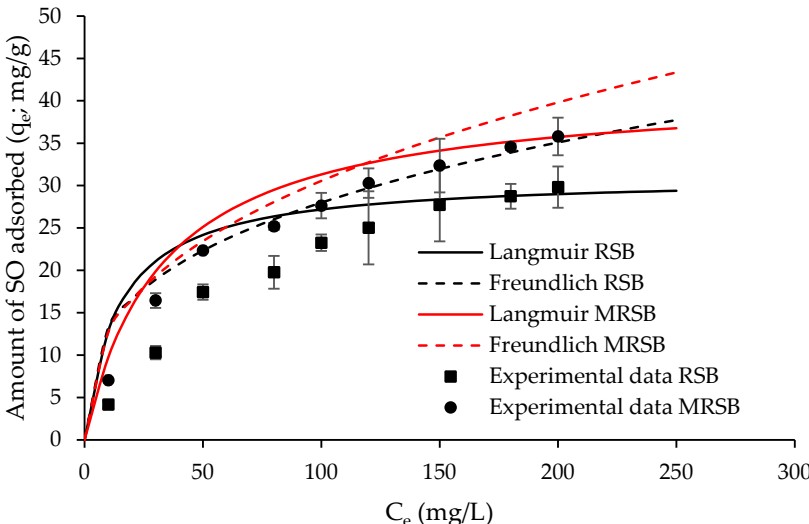

**Figure 5.** Langmuir and Freundlich isotherm of SO adsorption on RSB and MRSB (Experimental conditions: $C_0$ = 10–200 mg/L, $m_{adsorbent}$ = 2 g/L, t = 240 min, pH ~7).

The Langmuir and Freundlich isotherm parameters obtained from the adsorption experiments are tabulated in Table 2. It is obvious that the experimental data fit both the Langmuir and Freundlich isotherm models well. The Freundlich correlation coefficients were 0.99 for MRSB and 0.97 for RSB, while the Langmuir correlation coefficients were 0.98 for the MRSB and 0.96 for the RSB, respectively.

**Table 2.** The Langmuir and Freundlich isotherm parameters for SO adsorption onto MRSB and RSB.

| Adsorbent | Langmuir | | | Freundlich | | |
|---|---|---|---|---|---|---|
| | $q_m$ mg/g | $k_L$ L/mg | $R^2$ | $1/n$ | $k_F$ $(mg/kg)/(mg/L)^n$ | $R^2$ |
| RSB | 31.06 | 0.07 | 0.956 | 3.07 | 6.24 | 0.970 |
| MRSB | 41.59 | 0.03 | 0.977 | 2.62 | 5.26 | 0.989 |

Under the optimum adsorption conditions, the maximum adsorption capacity of RSB for SO was 31.06 mg/g, while that of MRSB for SO was 41.59 mg/g (Table 2), and the adsorption capacity of MRSB for SO is therefore close to 1.4 times greater than RSB. Several studies also demonstrated that with magnetic coating, the magnetic response and adsorption capacity of biochar to organic pollutants from the aqueous phase are enhanced, as listed in Table 3.

**Table 3.** Comparison of previous studies on the removal capacity of various contaminants between magnetic biochar and pristine biochar.

| Feedstock | Pyrolytic Temperature (°C) | Technique Used | Contaminants | Effect on Removal Capacity | Ref. |
|---|---|---|---|---|---|
| Corncob | 160 | Hydrothermal carbonization | Methylene blue | Near 1.6 times higher than pristine biochar | [14] |
| Corn stalks | 400 | Chemical co-precipitation | Crystal violet | Near 20 times higher than pristine biochar | [16] |
| Loblolly pine wood | 600 | Pyrolysis of hematite-treated biomass | As (V) | Near 1.6 times higher than pristine biochar | [24] |
| Rice husk | 400, 500 and 600 | Chemical co-precipitation | rhodamine 6G | Higher sorption capability than the pristine biochars | [25] |
| Orange peel | 250, 400 and 700 | Chemical co-precipitation | HOCs and phosphate | Higher sorption capability than the pristine biochars | [15] |
| Rice husk and the organic fraction of municipal solid wastes | 300 | The biomass was impregnated with calcium and iron agents before pyrolysis | As(V) and Cr (VI) | Much better As(V) removal capacity compared to the non-impregnated biochars | [26] |
| Rice straw | 500 | Chemical co-precipitation | Safranin O | Near 1.4 times higher than pristine biochar | This paper |

Overall, magnetic biochar derived from rice straw was proven to offer good Safranin O adsorption rates along with its superior magnetic properties. However, the present work attempted to produce biochar and magnetic biochar adsorbents from rice straw and assess their ability to remove a particular species or ion (i.e., Safranin O) in synthetic wastewater, in which SO dye solution is prepared and treated with adsorbents. In other words, in this study, the biochar and magnetic biochar derived from rice straw primarily performs their role in the context of single-ion Safranin O testing. In the case where they are applied for the assessment of complex matrices (i.e., whole industrial effluent testing or real textile wastewater), the evaluation of test ions must be done with caution to exclude

interferences from unintended species or ions. In addition, further work with real industrial-like textile or dyeing effluents should be considered for the removal of other contamination species or ions.

### 3.3. Adsorption Kinetic Analysis

It is necessary to estimate the adsorption rates and adsorption kinetics under experimental conditions [27]. In this study, the kinetic models used include a pseudo-first-order kinetic model and a pseudo-second-order kinetic model, and below are their non-linear model equations:

$$q_t = q_e(1 - \exp^{-k_1 t})$$

$$\frac{dq_t}{dt} = k_2(q_e - q_t)^2$$

where $q_e$ (mg/g) represents the equilibrium adsorption capacity; $q_t$ (mg/g) represents the t time adsorption capacity; $k_1$ (1/min) represents the constant rate of pseudo-first-order adsorption; and $k_2$ (g/mg·min) represents the constant rate of pseudo-second-order adsorption. Figure 6 shows the experimental data of RSB and MRSB toward SO fitted with the two kinetic models.

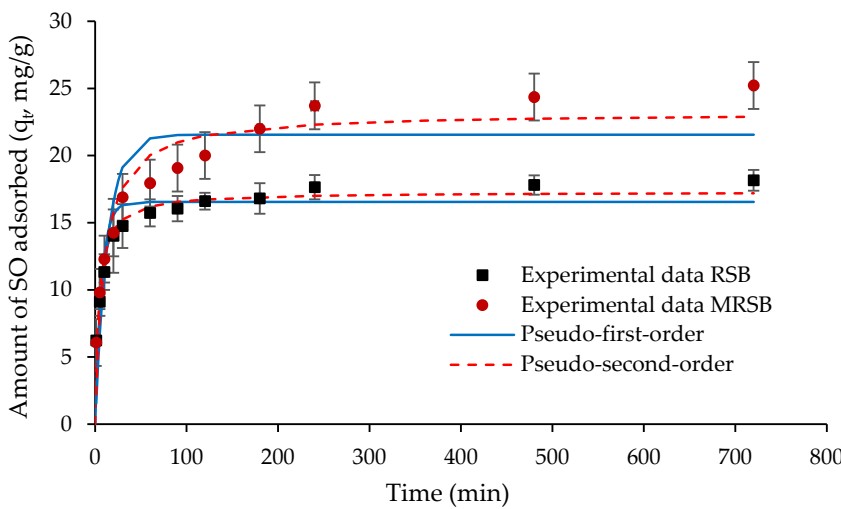

**Figure 6.** Fitting of pseudo-first-order and pseudo-second-order models for SO adsorption onto RSB and MRSB (Experimental conditions: $C_0$ = 50 mg/L, $m_{adsorbent}$ = 2g/L, t = 1–720 min, pH ~7).

The kinetic parameters for SO adsorption onto biochar are tabulated in Table 4. According to Table 4, the regression coefficient $R^2$ value of the pseudo-second-order kinetic equation (0.92 for RSB, 0.90 for MRSB) was greater than the values obtained by the pseudo-first-order kinetic equation (0.81 for RSB, 0.78 for MRSB). Moreover, from pseudo-second-order kinetics, the experimental values ($q_{e,exp}$) of both MRSB and RSB show better agreement with their calculated values of the equilibrium adsorption capacity ($q_{e,cal}$).

**Table 4.** Kinetic parameters of SO adsorption on RSB and MRSB.

| Adsorbent | Pseudo-First-Order | | | | Pseudo-Second-Order | | | |
|---|---|---|---|---|---|---|---|---|
| | $q_{e,exp}$ mg/g | $q_{e,cal}$ mg/g | $k_1$ 1/min | $R^2$ | $q_{e,exp}$ mg/g | $q_{e,cal}$ mg/g | $k_2$ g/mg·min | $R^2$ |
| RSB | 18.16 | 16.55 | 0.14 | 0.81 | 18.16 | 17.29 | 0.01 | 0.92 |
| MRSB | 25.22 | 21.55 | 0.07 | 0.78 | 25.22 | 23.20 | 0.00 | 0.90 |

In view of these results, it can be safely concluded that the kinetic process was better fitted to the pseudo-second-order kinetic model for both RSB and MRSB. In the adsorption process, the pseudo-second-order kinetic model implies chemisorption as the rate-determining step [28]. Other magnetic biochar studies on dye adsorption have also shown good fits to the pseudo-second-order kinetic model [29,30]. Chemisorption or chemical adsorption refers to the phenomenon of adsorbate molecules adhering to a solid adsorbent surface by strong chemical bonding, especially covalent bonds [31].

The kinetic fitted results already mentioned that chemical adsorption may be involved in the adsorption process of SO by RSB and MRSB. However, the pseudo-second-order kinetic model does not indicate that the adsorption process solely followed the chemisorption mechanism. Therefore, to further identify the rate-controlling step, the Weber-Morris intra-particle diffusion model was applied. The below equation represents the model:

$$q_t = k_{id}t^{1/2} + C,$$

where $q_t$ (mg/g) is the t time adsorption capacity; C is the intercept; and $k_{id}$ is the intra-particle diffusion rate constant. Figure 7 shows the intra-particle diffusion plots for adsorption capacity, and the parameters of the model are listed in Table 4.

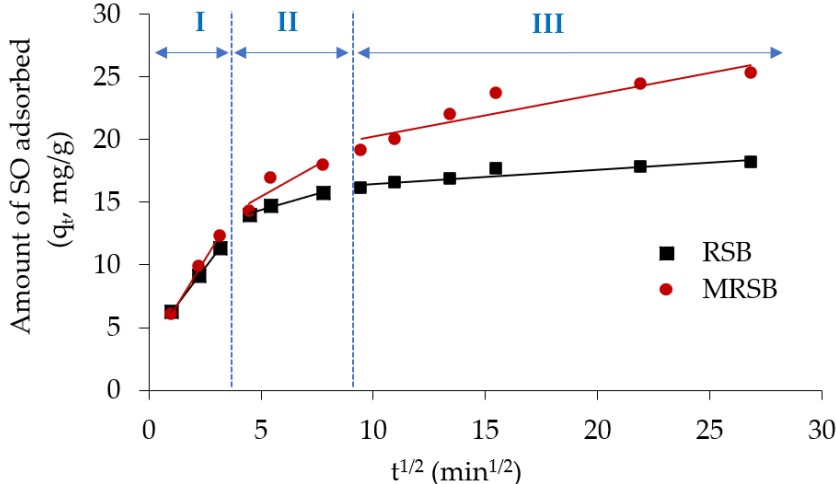

**Figure 7.** Intraparticle diffusion plots for adsorption of SO on MRSB and RSB (Experimental conditions: $C_0$ = 50 mg/L, $m_{adsorbent}$ = 2 g/L, t = 1–720 min, pH ~7).

The intra-particle diffusion graph shows that the SO adsorption onto the adsorbent materials was not linear over the whole experiment duration. In fact, the graph shows more than one kinetic stage in the SO adsorption process. The graph exhibited three slopes, and thus, three distinct stages of adsorption. The first stage (stage I, t = 1–10 min) is the external surface diffusion stage, involving the migration of the SO molecules from the solution to the fluid–solid interface (RSB's and MRSB's external surface). The second (stage II, t = 20–60 min) is the gradual adsorption stage, involving the transfer of the SO molecules into the pores of RSB and MRSB. The third stage (stage III, t > 60 min) is the equilibrium adsorption stage, where the speed of adsorption starts to slow down because of low SO concentration left in the solution, as well as less available adsorptive sites on the surfaces of RSB and MRSB. In the three stages, the second and third participated in the intra-particle diffusion process [32]. The values of $R^2$ for each stage were found to be in the range of 0.80 and 0.99, indicating that the adsorption process could be driven by an intraparticle diffusion mechanism. However, Figure 7 shows the straight lines of stages II

and III do not pass through the origin. The calculated $C_1$, $C_2$, $C_3$ values were also all non-zero (Table 5). This proves that the intra-particle diffusion is not the sole rate-limiting step.

**Table 5.** Intra-particle diffusion kinetic parameters of SO adsorption on RSB and MRSB.

| Adsorbent | Intra-Particle Diffusion | | | | | | | | |
|---|---|---|---|---|---|---|---|---|---|
| | Stage I: t = 1–10 min | | | Stage II: t = 20–60 min | | | Stage III: t > 60 min | | |
| | $C_1$ mg/g | $k_{id1}$ mg/g·min$^{1/2}$ | $R^2$ | $C_2$ mg/g | $k_{id2}$ mg/g·min$^{1/2}$ | $R^2$ | $C_3$ mg/g | $k_{id3}$ mg/g·min$^{1/2}$ | $R^2$ |
| RSB | 3.87 | 2.35 | 0.99 | 11.85 | 0.51 | 0.98 | 15.35 | 0.11 | 0.85 |
| MRSB | 3.24 | 2.88 | 0.99 | 10.37 | 1.02 | 0.80 | 16.89 | 0.34 | 0.84 |

It can be seen that the $k_{id1}$ was higher than the $k_{id2}$ and $k_{id3}$ (Table 5) for both RSB and MRSB materials. This demonstrates that the rate constant of the external surface adsorption ($k_{id1}$) was faster than the rate of the gradual adsorption stage ($k_{id2}$) and the equilibrium adsorption stage ($k_{id3}$) of the intra-particle diffusion. This also demonstrates that the diffusion resistance of the boundary layer was much smaller than the diffusion resistance of the pore diffusion steps; thus, external diffusion was necessarily involved in the adsorption process. Therefore, the entire adsorption kinetics of SO by both RSB and MRSB may be governed by external diffusion and intra-particle diffusion at the same time.

Overall, in this study, the mechanism governing SO adsorption on the RSB and MRSB may be controlled by both physisorption and chemisorption. Figure 8 shows the possible physical forces involved, including: (1) porous diffusion, controlling by both external diffusion and intra-particle diffusion governances; (2) H-bonding, where carboxyl (−COOH) and hydroxyl (−OH) groups of RSB/MRSB are bonded directly to a nitrogen atom of an −NH$_2$ group of SO molecules; (3) $\pi$-$\pi$ interaction, bonding between the aromatic rings of RSB/MRSB and of SO molecules; and (4) $\pi^+$-$\pi$ interaction, bonding between cation N$^+$ of the SO molecules and the aromatic rings of RSB/MRSB. In contrast, for chemisorption, an interaction between the negatively charged surface (i.e., carboxyl −COOH and hydroxyl −OH groups) of RSB/MRSB and positively charged (i.e., N$^+$) SO dye takes place. Therefore, the collective physical and chemical forces account for the adsorption mechanism of SO dye molecules by both RSB and MRSB adsorbents.

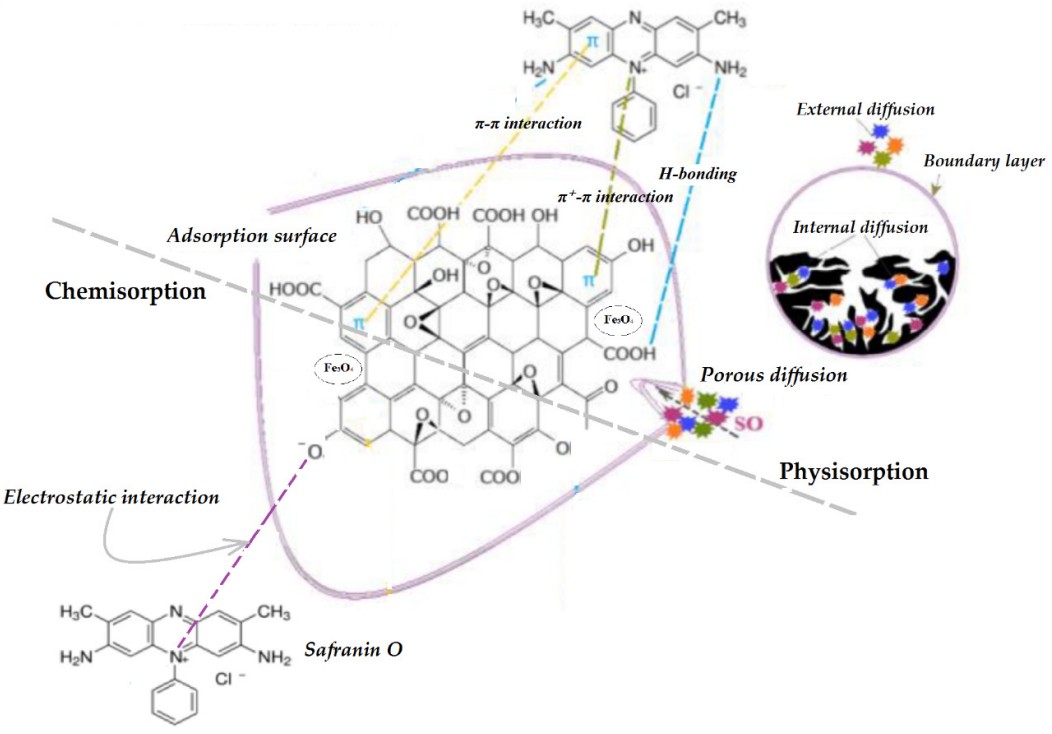

**Figure 8.** Depicting the possible adsorption mechanism of SO on both RSB and MRSB.

The ability to collect magnetic rice straw biochar after being used for SO adsorption from the solution is shown in Figure 9. An external magnet rapidly collected all the MRSB introduced into the suspension of MRSB- and SO-contaminated water. Therefore, MRSB adsorbent can be easily separated from water after adsorption by a magnet.

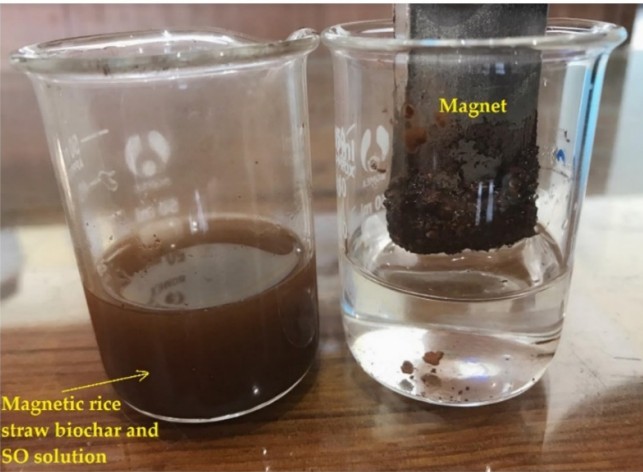

**Figure 9.** Introduction of an external magnet into the suspension of MRSB- and SO-contaminated water.

## 4. Conclusions

This study explored the physicochemical characteristics and comparatively evaluated Safranin O adsorption capacity between biochar and magnetic biochar from rice straw using batch experiments. Our results suggest that magnetic biochar improved adsorption properties over non-magnetic biochar, since magnetization helps in improving

the specific surface area, increasing the number of functional groups, all which may provide more SO adsorption sites. The theoretical maximum adsorption capacities of the biochar and magnetic biochar were 31.06 and 41.59 mg SO per gram of dry adsorbent, respectively. The adsorption capacity of magnetic biochar for SO is therefore close to 1.4 times greater than its precursor. Their experimental data are well-described by both Freundlich and Langmuir isotherms, with correlation coefficients higher than 0.96. In addition, their kinetic data fit well with the pseudo-second-order kinetic model, as well as the intra-particle diffusion model, which suggested that the collective physical and chemical forces may account for the adsorption mechanism of Safranin O molecules by both adsorbents, including porous diffusion, H-bonding, the $\pi$-$\pi$ interaction, $\pi^+$-$\pi$ interaction, and electrostatic attraction. Under the action of a magnetic field, the magnetic biochar material is easily collected from water after adsorption.

From an industrial point of view, the removal of pollutants including dye ions using continuous flow systems is found very useful and reliable. Therefore, it is necessary to examine the practical applicability of the biochar/magnetic biochar adsorbents from this work for real-world textile wastewater in the continuous-flow mode.

Regeneration of an adsorbent is of crucial importance in industrial practices for the removal of dye pollutants from textile wastewaters. Regeneration generally provides useful information to allow for the economic design of an overall operation and to make the adsorption process more feasible and practical. Therefore, regeneration studies and desorption modelling steps should be considered in future work.

**Supplementary Materials:** The following are available online at https://www.mdpi.com/article/10.3390/w14020186/s1, Figure S1: SEM image of RSB, Figure S2: SEM image of MRSB, Figure S3: Effect of pH, Figure S4: Effect of RSB and MRSB dosage, Figure S5: Effect of initial SO concentration, Figure S6: Effect of contact time.

**Author Contributions:** All authors have contributed equally. All authors have read and agreed to the published version of the manuscript.

**Funding:** This study is funded in part by the Can Tho University, grant number T2021-67. The APC was funded by authors's budget.

**Institutional Review Board Statement:** Not applicable.

**Informed Consent Statement:** Not applicable.

**Data Availability Statement:** Not applicable.

**Acknowledgments:** The authors are extremely grateful to Nigel K. Downes for his correction of grammatical errors and English editing. The authors also gratefully acknowledge Phan Thi Thanh Tuyen, Kieu Thi Khanh, Pham Thi Ngoc Tran, Pham Thi Chuc Lan, Nguyen Hoang Son, Pham Thai Sang for their assistance with the experiment.

**Conflicts of Interest:** The authors declare no conflict of interest.

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
