# Peer review of "Rice Straw Biochar and Magnetic Rice Straw Biochar for Safranin O Adsorption from Aqueous Solution"

_water, doi:10.3390/w14020186_

Round 1

Reviewer 1 Report

Very good presented work, just to finalise for publish , you can provide a table for comparison to similar works.

-Other heat treatment method can be used for biochar development and magnetisation. So, you can check/use the following for discussion part especially in removal efficiency:

https://www.sciencedirect.com/science/article/pii/S2090123221001478

Author Response

Dear Reviewer,

We are most thankful for your thoughtful comments on our manuscript. We also would like to acknowledge the time and effort that the reviewer has invested into assessing the previous version of our manuscript. You have raised important issues and provided helpful inputs for improving our manuscript. We agree with all your comments and have revised our manuscript accordingly.

The reviewers’ comments are repeated in italics and our responses are inserted directly after each comment in the attaching document. We hope that the reviewer will find our responses satisfactory.

Looking forward hearing from you soon.

Yours sincerely,

Reviewer 2 Report

The authors reported the development of rice straw biochar and magnetic rice straw biochar and their application for the removal of Safranin O. The characterization aspect of the prepared materials was sufficient, and the practical aspect (removal of Safranin O) was reasonably and sufficiently studied. However, the paper is lacking in shedding more light on the role of magnetite in either the adsorption process or recovery. The method of disposal for spent adsorbent was not discussed. This manuscript needs major revision and I have listed these issues and recommendations in chronological order. Following is a summary of the major corrections and revisions:

  1. The introduction needs to put forward the highlights of your research. Please follow the literature review and show the knowledge gaps identified and link them to your research objectives. I strongly suggest the authors to add more recently references.
  2. Line 138: Details regarding the sample preparation for SEM analysis should also be added. Regarding the FTIR measurements, it should be important to mention the scans used and also the resolution. These details are important for the readers.
  3. Line 228: Please do not use linearization of the isotherm equations (the same for kinetic equations). Nowadays, most computer programs can perform non-linear regression and should be used in preference of linearization to determine adsorption parameters. Comment please.
  4. Line 348: Figure 9 is missing.
  5. Conclusions: Conclusions need to be improved by specifying the discussed important points within this work. In the conclusions, the author should also provide an outlook of the challenges and potential future directions.

General comments:

  • The biggest concern with the approach presented here is the potential for sorption of unintended species in the water. Reasonably capacity is demonstrated, which is important, but selectivity is at least as important as capacity. The authors show that the sorbent can capture Safranin O, but this study does not evaluate how many interfering species are also captured, which will limit the practical effectiveness. This topic should be addressed explicitly in the discussion.
  • The major drawback of this paper is followed by several questions: Can this work be feasible to be done in industrial scale, and can it be scaled up? What is the novelty of this work in context of related composites used for the removal of other similar effluents compositions?
  • Can the same experiments be done using continuous adsorption column?
  • Can this method be scaled up? Cite updated papers in the said query, include it in the introduction, and conclusion part of your revised Manuscript.
  • In order to make the adsorption process more feasible, the adsorbent is usually regenerated. Data regarding the recycling performance of the material should be added in the manuscript. Moreover, please provide the XRD, SEM, and FTIR results of the adsorbent after the five times of regeneration.

Author Response

(The authors gave the same response as above.)

Reviewer 3 Report

The paper presents research on adsorption from aqueous solution using rice straw biochar and magnetic rice straw biochar. The presentation of methods and scientific results in the current form is satisfactory for publication in Water journal. The paper is interesting, but unfortunately, a bit is poorly written. For this reason, I would like to see the next version of the article again before the final acceptance. The minor and significant drawbacks to be addressed can be specified as follows:
1.    Tittle. straw ---> Straw.
2.    Abstract, BET specific surface areas and pHpzc – these are characterization methods (numerical values), not experimental techniques. “BET specific surface areas and pHpzc” ---> “N2 adsorption (77K) and point of zero charge measurements”.
3.    Line 74. The equation is not balanced in terms of mass and charge!!! Fe3+ ---> 2 Fe3+
4.    Line 78, “BET specific surface are”. See point no. 2.
5.    Line 138. Japan)) ---> Japan).
6.    Line 144 – 146. Low-temperature nitrogen adsorption?
7.    Line 152, HS250 BASIC, IKA LABORTECHNIK. Caps Lock?Line 168, 
8.    Line 168. measurement of BET specific surface area ---> calculation of BET specific surface area
9.    Tab. 1. I miss nitrogen adsorption isotherms for the tested materials. What type of materials are they - microporous or mesoporous?
10.    Fig. 2. Results for RSB should appear at the top. First (A) then (B).
11.    Eq. (3). Please reject “(Freudlich model)”.
12.    Fig. 5. One point is missing for RSB (Ce=175mg/L).
13.    Line 250. R is the correlation coefficient. correlation coefficients ---> determination coefficients.
14.    Tab. 2. (i) RMSB should be bold contrary to qm, mg/g, and 31.06. See also Tab. 3 (ii) the values of R2 should be given with an accuracy of 3-4 decimal places – see also Tab. 3 (iii) “(mg/kg)/(mg/L)n” n?
15.    Eqs. (4) and (5). “(pseudo-first-order model)” and “(pseudo-second-order model)” should be rejected.
16.    Fig. 9. The authors did not include this figure !!!!!
17.    Line 354. See point no. 2.
18.    Fig. 1S and 2S. It should be the exact resolution scale.

Author Response

(The authors gave the same response as above.)

Round 2

Reviewer 2 Report

The authors have addressed my queries satisfactorily. Manuscript was improved in accordance to my suggestions and I have no further objection to this paper. The quality of the paper has been improved and, therefore, I consider that the article can be accepted in present form.

Author Response

Once again, we would like to acknowledge the time and effort that the reviewer has thoroughly revised our work.

Reviewer 3 Report

In my opinion, the corrected work can be accepted for publication — however, minor revision.

However, I have minor comments.
1.    Supplementary materials. “were measured via the BET method,” ---> “were calculated via the BET method,”
2.    Line 16. “BET specific surface areas and pHpzc measurements.” ---> “N2 adsorption (77K) and pHpzc measurements.” In the case of N2 - 2 as a subscript.
3.    Lines 145 and 146. “The BET specific surface area was measured using Nova Station A” ----> “The BET specific surface area was calculated on the basis of low-temperature nitrogen adsorption isotherm measured using Nova Station A”.
4.    “The nitrogen adsorption isotherms we obtained from BET analysis results of both”, Response to comments from Referee #3. The authors must consult a specialist in this field. Adsorption isotherms are measured. On the basis of these data, the apparent surface areas are determined using the BET method. I have a huge request to the authors for a thorough review of the work again because I would be ashamed if such things appeared in the final version of the article.

Author Response

Dear Reviewer,

We honestly would like to acknowledge the time and effort that the reviewer has thoroughly revised and corrected our manuscript.

Yours sincerely,

This manuscript is a resubmission of an earlier submission. The following is a list of the peer review reports and author responses from that submission.